# Electronic paddle-wheels in a solid-state electrolyte

Harender S. Dhattarwal [1], Rahul Somni[1] & Richard C. Remsing [1] ✉

Solid-state superionic conductors (SSICs) are promising alternatives to liquid electrolytes in batteries and other energy storage technologies. The rational design of SSICs and ultimately their deployment in battery technologies is hindered by the lack of a thorough understanding of their ion conduction mechanisms. In SSICs containing molecular ions, rotational dynamics couple with translational diffusion to create a paddle-wheel effect that facilitates conduction. The paddle-wheel mechanism explains many important features of molecular SSICs, but an explanation for ion conduction and anharmonic lattice dynamics in SSICs composed of monatomic ions is still needed. We predict that ion conduction in the classic SSIC AgI involves electronic paddle-wheels, rotational motion of localized electron pairs that couples to and facilitates ion diffusion. The electronic paddle-wheel mechanism creates a universal perspective for understanding ion conductivity in both monatomic and molecular SSICs that will create design principles for engineering solid-state electrolytes from the electronic level up to the macroscale.

Solid-state superionic conductors (SSICs) promise significant improvements over liquid electrolytes for energy storage technologies[1–4]. Because of their use as electrolytes in solid-state batteries, SSICs have been studied extensively over the past 50 years in an effort to understand the molecular-scale principles underlying conduction[5–7]. Conduction in molecular ionic solids is well understood. In solids like lithium sulfate, $Li^+$ cations rapidly diffuse through the rigid lattice made by molecular $SO_4^{2-}$ anions, with rotational motion of sulfate often facilitating the rapid cationic diffusion[6,8–12]. This phenomenon of the rotational motion of a molecular ion accompanying the conductivity of another ion in orientationally disordered molecular solids is termed the paddle-wheel effect. The reorientation of molecular ions leads to fluctuations in the local potential that affect the diffusion of mobile ions[13]. This paddle-wheel motion has since become a principle for designing molecular SSICs[11,14]. In contrast, similar principles have not been identified for SSICs composed of monatomic ions, like the prototypical AgI[6], where anionic lattice relaxations are connected to the diffusion of cations[15]. Fundamental questions concerning the coupling between the mobile cations and the immobile anionic lattice still remain[15,16].

In this work, we provide insight into the coupled cation-anion dynamics in SSICs by identifying an electronic analog of paddle-wheel motion in AgI. Our investigation into AgI is motivated by recent ideas concerning hidden disorder in otherwise ordered materials. This hidden disorder is typically local and electronic in nature, arising from an interplay between electronic and nuclear degrees of freedom[17]. We exploit the concept of electronic plastic crystals[18,19]—phases of crystalline solids in which localized lone pair electrons are dynamically orientationally disordered—to identify and characterize electronic paddle-wheels that accompany ionic diffusion in AgI. These insights help move one step closer to a unified framework for understanding ion transport in monatomic SSICs in line with molecular liquid and solid electrolytes[20].

## Results

In AgI, localized electron pairs exist on the iodide ions, and we find that AgI in its superionic phase is an electronic plastic crystal. The rotational motion of $I^-$ lone pair electrons accompanies the diffusion of $Ag^+$ through the SSIC, creating electronic paddle-wheels. As summarized by the simulation snapshots in Fig. 1a, electronic paddle-wheels in AgI are defined by the rotation of $I^-$ lone pairs (light purple) facilitating the motion of an $Ag^+$ cation (orange) from one site to another.

Before quantifying dynamics in AgI, we first characterize the structural correlations involving lone pairs to identify cation

[1]Department of Chemistry and Chemical Biology, Rutgers University, Piscataway, NJ, USA. ✉e-mail: rick.remsing@rutgers.edu

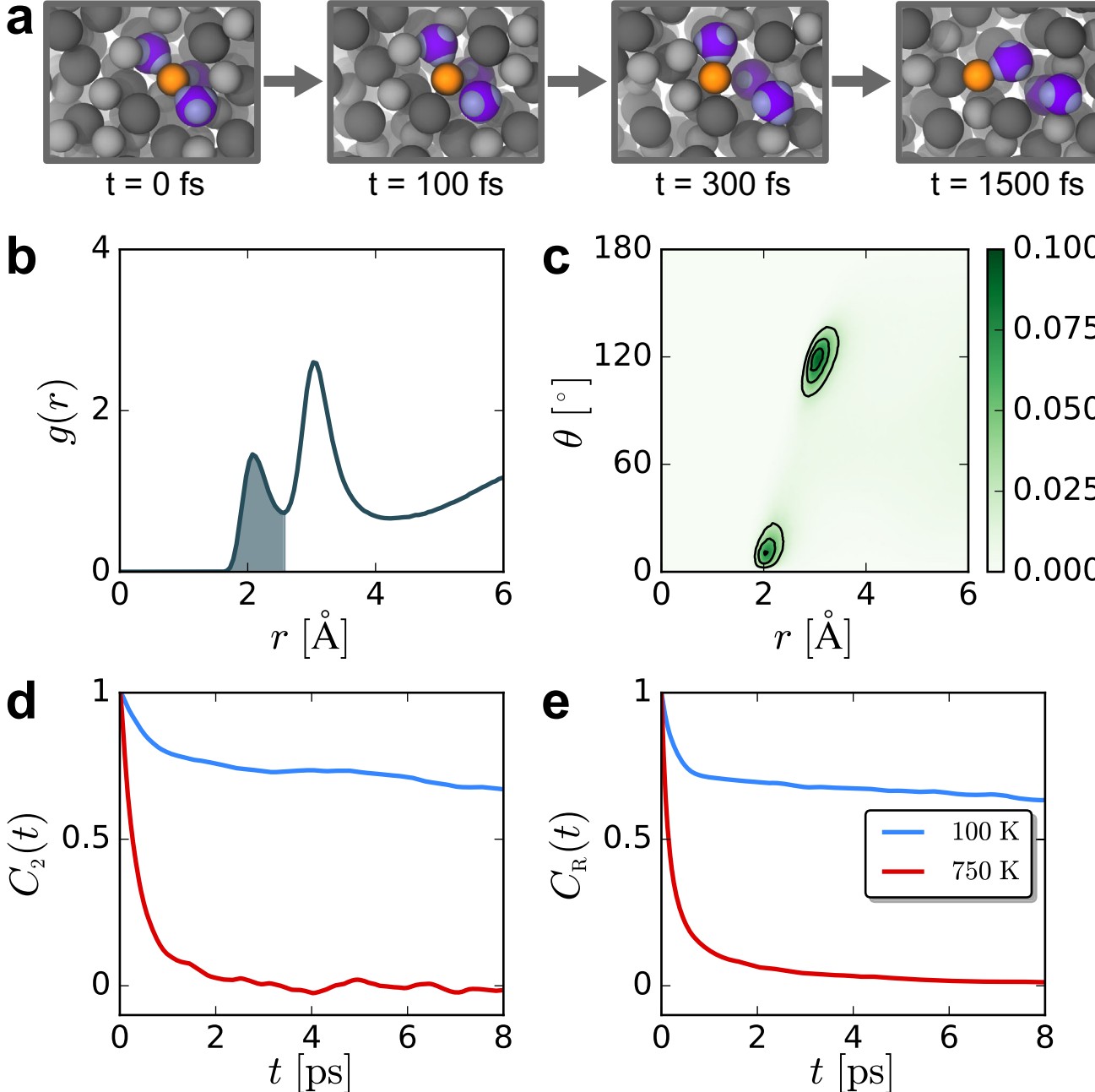

**Fig. 1 | Anion lone pair rotations accompany cation diffusion. a** Snapshots from a trajectory of AgI at 750 K illustrating the electronic paddle-wheel motion that facilitates $Ag^+$ conduction. A single $Ag^+$ (orange sphere) and its initial three closest $I^-$ (purple spheres) with their lone pairs, represented as maximally localized Wannier function centers (MLWFCs) (small, light purple spheres) are highlighted. All other $Ag^+$ and $I^-$ are shown as small gray and larger dark gray spheres. The lone pairs rotate concertedly with the motion of $Ag^+$ as it moves from one binding site to the next, moving from left to right along the panels shown here. Following one rotation over a 300 fs time interval, the $Ag^+$ stays in a binding site until it diffuses with another lone pair rotation, with the waiting time and rotation time totaling approximately 1200 fs. **b** Radial distribution functions, $g(r)$, between $Ag^+$ cations and iodide lone pair MLWFCs. The integration of the first peak (shaded region) is proportional to the coordination number of lone pairs around $Ag^+$ ions. **c** Joint probability distribution of the lone pair-$Ag^+$ distance ($r$) and the $I^-$-lone pair-$Ag^+$ angle ($\theta$). **d** Tetrahedral rotor function time correlation functions (TCFs), $C_2(t)$, characterizing $I^-$ lone pair rotational dynamics. **e** $Ag^+$ cage TCF, $C_R(t)$, quantifying the diffusion of cations between binding sites.

coordination environments. We focus on the radial pair distribution function, $g(r)$, quantifying two-body correlations among cations, anions, and iodide lone pairs. We represent the positions of lone pairs using maximally localized Wannier function centers (MLWFCs)[21], which give a reasonable description of the location of localized electron pairs. The $Ag^+$-lone pair $g(r)$ exhibits two peaks at short distances, Fig. 1b. The first peak corresponds to close, direct interactions between lone pairs and $Ag^+$. Integration of this peak (shaded region) yields a

coordination number between two and three, indicating that each $Ag^+$ is coordinated on average by 2 or 3 sets of lone pairs. The second peak between 3 and 4 Å corresponds to correlations between the central $Ag^+$ and lone pairs on the iodide that are pointing away from the cation.

The coordination structure suggested by $g(r)$ is further supported through the joint probability distribution of the lone pair-$Ag^+$ distance ($r$) and the $I^-$-lone pair-$Ag^+$ angle ($\theta$), Fig. 1c. The peak near 2 Å in $g(r)$ involves linear $I^-$-lone pair-$Ag^+$ arrangements, consistent with direct

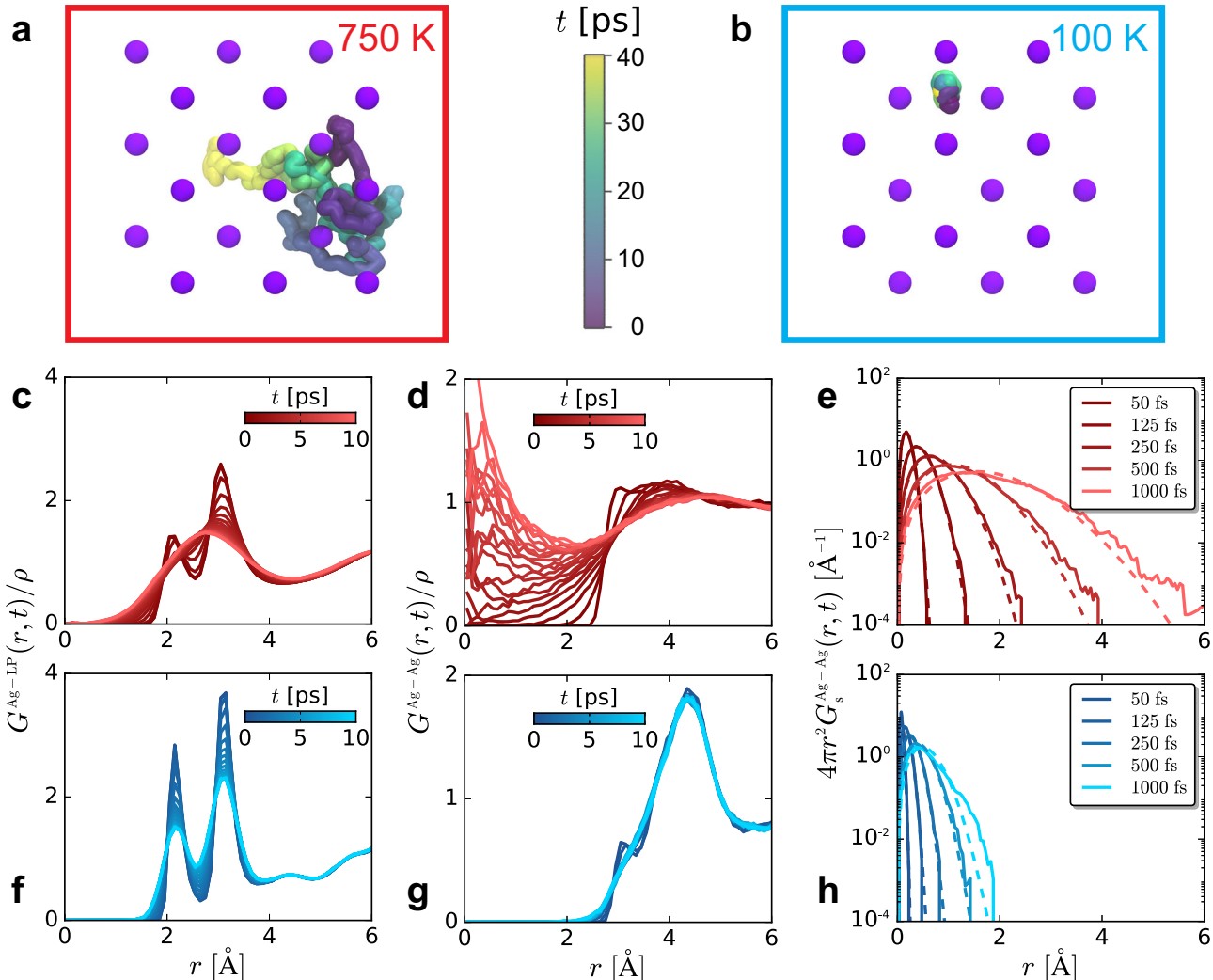

**Fig. 2 | Cation dynamics are collective and diffusive.** Snapshots showing the trajectory of a single Ag$^+$ at **a** 750 K and **b** 100 K, where the Ag$^+$ is colored by time according to the color bar, and iodides are shown as purple spheres. The space-time dynamics in the system are quantified through the distinct van Hove correlation function for cation-lone pair correlations at **c** 750 K and **f** 100 K, the distinct van Hove correlation function for cation-cation correlations at **d** 750 K and **g** 100 K, all normalized by the bulk density, $\rho$, and the self van Hove correlation function for cation correlations at **e** 750 K and **h** 100 K. The color palettes of the plots in **c**–**h** reflect the temperatures of the systems in **a**, **b**.

coordination of the cation by lone pair electrons. The peak near 3 Å involves values of $\theta \approx 120°$, consistent with the three other tetrahedral lone pair sites of I$^-$ pointing away from the Ag$^+$. This structural analysis strongly suggests that lone pair electrons are coordinating silver cations in AgI.

In an electronic plastic crystal, lone pair electrons exhibit dynamic orientational disorder. To quantify the rotational dynamics of I$^-$ lone pairs, we follow previous work and focus on tetrahedral rotor functions, $M_\gamma$, of order $l = 3$, such that $\gamma$ labels the $(2l + 1)$ functions for each $l$[18,19,22]. The tetrahedral rotor functions do not distinguish between the four I$^-$ MLWFCs and treat them all identically, as desired for identical electron pairs. We focus on the $\gamma = 2$ rotor function here, $M_2 = \frac{3\sqrt{5}}{40}\sum_{i=1}^{4}(5x_i^3 - 3x_i r_i^2)$, where $\mathbf{r}_i = (x_i, y_i, z_i)$ is a unit vector along I-MLWFC bond $i$, and $r_i = |\mathbf{r}_i|$. The rotational dynamics of I$^-$ lone pairs can be quantified through the time correlation function

$$C_\gamma(t) = \frac{\langle M_\gamma(0)M_\gamma(t)\rangle}{\langle M_\gamma^2(0)\rangle}, \qquad (1)$$

where $\langle \cdots \rangle$ indicates an ensemble average. In $\alpha$-AgI at 750 K, above the superionic transition at 420 K[6], the rotational TCF $C_2(t)$ decays with a relaxation time of 0.4–0.5 ps, indicating that MLWFC rotational motion occurs on a picosecond timescale, Fig. 1d. In comparison, similar MLWFC rotations in cesium tin halide perovskites occur on faster timescales of 0.04–0.1 ps[18,19].

When Ag$^+$ diffuses between the various sites in the AgI lattice, the identity of the iodides making up its coordination cage changes. Therefore, we quantify Ag$^+$ dynamics through a TCF that probes these changes in coordination cage, inspired by the cage correlation functions used in the study of supercooled liquids[23]. We define two indicator functions, $h_i^{\text{out}}(0,t) = \Theta(1 - \xi_i^{\text{out}}(0,t))$ and $h_i^{\text{in}}(0,t) = \Theta(1 - \xi_i^{\text{in}}(0,t))$, where $\Theta(x)$ is the Heaviside step function,

$$\xi_i^{\text{out}}(0,t) = \frac{l_i(0) \cdot l_i(t)}{l_i(0) \cdot l_i(0)}, \qquad (2)$$

$$\xi_i^{\text{in}}(0,t) = \frac{l_i(0) \cdot l_i(t)}{l_i(t) \cdot l_i(t)}, \qquad (3)$$

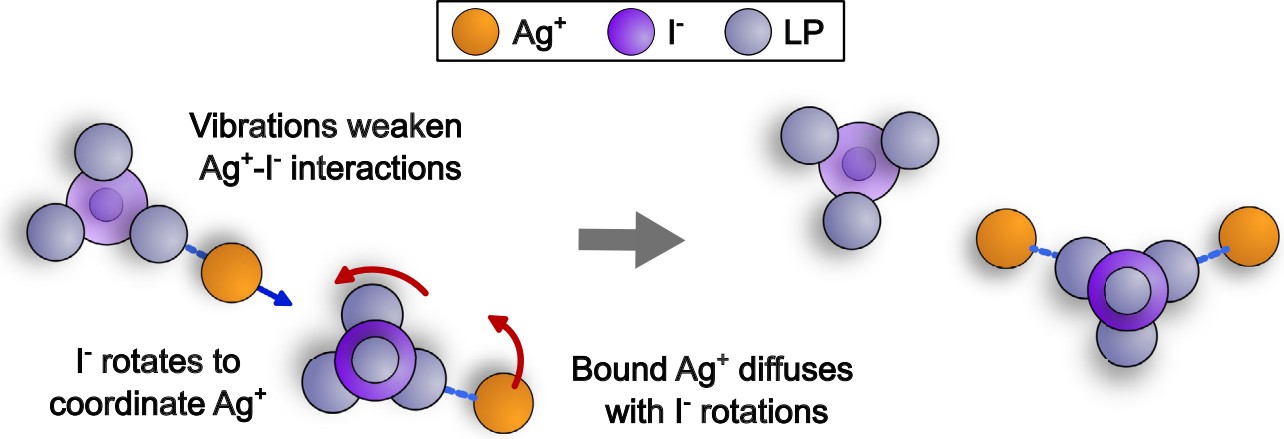

**Fig. 3 | Mechanism of ion diffusion with electronic paddle-wheels.** Schematic diagram of the collective diffusion mechanism of Ag⁺ in AgI. Thermal vibrational motion weakens some cation-anion interactions, new cation-anion interactions are formed through lone pair (LP) rotation, and these lone pair rotations are coupled to the diffusion of a coordinated Ag⁺.

and $l_i(t)$ is the neighbor list of the $i$th Ag⁺ at time $t$. Therefore, $h_i^{\mathrm{out/in}}(0,t) = 1$ if the neighbors are the same at times $t$ and 0. If the number of neighbors decreases, then $h_i^{\mathrm{out}}(0,t) = 0$ and $h_i^{\mathrm{in}}(0,t) = 1$. If the number of neighbors increases, then $h_i^{\mathrm{in}}(0,t) = 0$ and $h_i^{\mathrm{out}}(0,t) = 1$. We then define a cage TCF as $C_R(t) = \langle h_i^{\mathrm{out}}(0,t) h_i^{\mathrm{in}}(0,t) \rangle$, which quantifies the diffusion of Ag⁺ between occupation sites through changes in its coordination cage, like the motion shown in Fig. 1a. The cage TCF has a relaxation time of approximately 0.6–0.7 ps at 750 K, Fig. 1e, which is on a similar timescale to lone pair rotations. The similar timescales for Ag⁺ diffusion out of a binding site and lone pair rotations suggests a coupling between electron rotational motion and cation diffusion.

The MLWFC rotational dynamics are temperature dependent, Fig. 1d. When supercooled to the low temperature of 100 K, where $\alpha$-AgI is no longer conductive (see Supplementary Figs. 1–3), lone pair rotational dynamics are significantly slowed and $\alpha$-AgI is no longer in an electronic plastic crystal phase. As a result, the lone pair rotation TCF, $C_2(t)$, and the TCF quantifying Ag⁺ diffusion, $C_R(t)$, both decay very little on the timescale of the simulations. This slowing of lone pair rotations and Ag⁺ diffusion upon cooling suggests that they are intimately connected; Ag⁺ cannot diffuse unless I⁻ lone pairs rotate.

To characterize cation diffusion in more detail, we computed the distinct and self parts of the van Hove correlation function, $G_d(r,t)$ and $G_s(r,t)$, respectively, defined according to

$$G_d(r,t) = \left\langle \frac{1}{N} \sum_i \sum_{j \neq i} \delta\left( \left| \mathbf{r}_i(0) - \mathbf{r}_j(t) \right| \right) \right\rangle \quad (4)$$

and

$$G_s(r,t) = \left\langle \frac{1}{N} \sum_i \delta\left( \left| \mathbf{r}_i(0) - \mathbf{r}_i(t) \right| \right) \right\rangle. \quad (5)$$

The distinct van Hove function, $G_d(r,t)$, quantifies space-time correlations of the initial position of one particle ($i$) with other particles in the system ($j$). The self van Hove function, $G_s(r,t)$, quantifies space-time correlations of a particle with its initial position. For $t = 0$, $G_d(r,0)$ is equal to the $g(r)$ between the two particle types of interest, and $G_s(r,0)$ is a delta function at $r = 0$.

The distinct van Hove function for cation-lone pair correlations suggests electronic paddle-wheels in the SSIC phase of AgI, Fig. 2c, f. At $t = 0$, $G_d(r,t)$ is the Ag⁺-MLWFC pair distribution function. As $t$ increases at $T = 750$ K, the first peak in $G_d(r,t)$ broadens on the timescale of paddle-wheel motion, indicating that Ag⁺ and the MLWFCs are

correlated. This correlation in cation and lone pair dynamics is reflective of electronic paddle-wheels. At low temperatures, when AgI is not a SSIC, $G_d(r,t)$ hardly changes shape, reflecting a lack of cation diffusion and anion lone pair rotation.

Cation motion in traditional molecular SSICs often occurs through discrete hops induced by paddle-wheel motion of molecular anions. Cations rattle in a binding site and then very rapidly move to another site during a hop. In contrast, we find that Ag⁺ dynamics are diffusive in AgI. The Ag⁺ trajectory shown in Fig. 2a shows no evidence of the rapid jumps indicative of hopping; the trajectory is smooth and diffusive.

The diffusive nature of cation dynamics in AgI at 750 K is further supported by $G_d(r,t)$ and $G_s(r,t)$ shown in Fig. 2d, e, while a lack of dynamics is suggested by those for AgI at 100 K, Fig. 2g, h. The trajectory shown in Fig. 2b corroborates this lack of dynamics in Ag⁺ ions at 100 K. At $t = 0$, the cationic $G_d(r,t)$ is equal to the Ag-Ag $g(r)$. As time progresses, a peak smoothly grows in near $r = 0$, the initial location of an Ag⁺. This peak at the origin at later times is consistent with a cation leaving its initial location and another cation taking its place. The smooth growth of this peak with time suggests a lack of hopping-like motion[24]. A similar lack of hopping can be observed in $G_s(r,t)$, Fig. 2e. Hopping manifests in $G_s(r,t)$ through the appearance of well-defined peaks at typical hopping distances. Peaks are not observed in the cation $G_s(r,t)$, and we instead find smooth distributions. However, the motion is not purely diffusive, which would result in the dashed Gaussian distributions. The deviations from Gaussian diffusion suggest that the dynamics are collective and dynamically heterogeneous, with some ions moving slower and faster than Gaussian expectations. Collective cation dynamics can be expected because a cation cannot diffuse to another binding site unless another cation leaves that site.

The mechanism of collective dynamics in AgI is suggested by our simulations, and it involves a combination of translational and rotational dynamics. We propose three main contributions that may occur nearly simultaneously, shown in Fig. 3. Thermally induced translational fluctuations weaken the strength of Ag⁺-I⁻ interaction. The Ag⁺ is then coordinated by a different I⁻, and the newly coordinating I⁻ rotates its lone pairs to form this new interaction. The rotating I⁻ coordinates other cations, and one of these coordinated cations diffuses by accompanying the lone pair rotation. Vibrational relaxations in the I⁻ sublattice make Ag⁺ more susceptible to diffusion[15]. This diffusion, however, is facilitated by I⁻ lone pair rotations.

The diffusive dynamics of the Ag⁺ cations observed here represent a fundamental difference between electronic paddle-wheels and molecular paddle-wheels in SSICs. Cation motion in molecular

paddle-wheels often occurs through hopping, but the motion is diffusive in electronic paddle-wheels. Despite this difference, we expect the analogy between electronic and molecular paddle-wheels to be useful, such that many of the concepts developed for molecular paddle-wheels may be extended to electronic paddle-wheels in monatomic SSICs.

## Discussion

To summarize, our molecular simulations suggest that the SSIC AgI exhibits hidden electronic disorder, and this dynamic electronic disorder leads to electronic paddle-wheels—anion lone pair rotations that couple to cation diffusion. Electronic paddle-wheels may explain important mysteries about conduction mechanisms of SSICs composed of monatomic ions. For example, collective Ag$^+$ diffusion was recently identified using topological analysis[25]. Our proposed diffusion mechanism involving lone pair rotations is consistent with this collective diffusion and provides an electronic origin for these collective dynamics. We also anticipate that the cation-anion coupling due to electronic paddle-wheels may be responsible for the anharmonic coupling observed in spectroscopy[15], similar to anharmonic couplings observed in molecular plastic crystals[26].

The ionic diffusion mechanism predicted here is analogous to the paddle-wheel motion observed in molecular solid-state electrolytes. We expect that this analogy can be leveraged to adapt our understanding of molecular electrolytes to monatomic electrolytes. As a result, electronic paddle-wheels expand unifying frameworks for crystalline, amorphous, and liquid molecular electrolytes to include SSICs composed of monatomic ions[20]. Similarly, this mechanism naturally expands the concept of frustration in SSICs to include dynamical electronic frustration resulting from lone pair rotations, in analogy to the dynamical frustration caused by molecular paddle-wheels[27,28]. As a result, we expect that hidden rotational electronic disorder and the resulting electronic paddle-wheels may be an important marker for designing electrolytes through tuning lone pair-mobile ion interactions, for example. We expect that electronic paddle-wheels are not limited to AgI but will be found in many SSICs. For example, while this work was in revision, a preprint appeared that suggested that tin ion lone pair rotations may impact the diffusion of fluoride ions in BaSnF$_4$[29], further emphasizing the importance of understanding electronic effects in SSICs. Similar to molecular paddle-wheels[11,30,31], understanding electronic paddle-wheels will create rational, electronic-scale design principles that leverage the coupling between lone pair rotations and cation diffusion to discover new superionic conductors.

## Methods

We performed Born-Oppenheimer molecular dynamics simulations with Kohn-Sham density functional theory using the CP2K software package version 7 employing a hybrid Gaussian and plane wave (GPW) approach for representing electron density[32]. By doing so, we neglect the fast electronic modes that require a detailed description of quantum dynamics, and instead focus on the slowest electronic modes that are coupled to the motion of the nuclei and most relevant to ion conduction. We employed the molecularly optimized double-$\zeta$ with valence polarization (DZVP) basis set[33] with cutoffs of 450 Ry for plane wave energy and 60 Ry for the reference grid. Goedecker-Teter-Hutter pseudopotentials compatible with employed basis sets were used to represent core electrons[34,35], while the valence electrons were represented explicitly using the Perdew-Burke-Ernzerhof (PBE) generalized gradient approximation to the exchange-correlation functional[36]. The **k**-point sampling was performed only at the Γ-point. Long-range dispersion interactions were included using Grimme's D3 van der Waals correction[37]. Our simulations used a $3 \times 3 \times 3$ supercell of a $\alpha$-AgI unit cell with edge length 5.0855 Å[6]. Starting structures were equilibrated for at least 50 ps at 100 K (supercooled $\alpha$-AgI) and 750 K in the

canonical ensemble (NVT) using the canonical velocity rescaling (CSVR) thermostat[38]. For the calculation of dynamic properties, equations of motion were propagated for 80 ps in the microcanonical ensemble using velocity Verlet integrator with a timestep of 0.5 fs. The last 40 ps of these trajectories were used for the analysis. Calculations for dynamic properties are averaged over multiple time origins. Maximally localized Wannier functions were obtained by minimizing their spreads within CP2K[21,39].

### Reporting summary

Further information on research design is available in the Nature Portfolio Reporting Summary linked to this article.

## Data availability

The data generated in this study are presented in the main article and the Source Data file. Source data are provided with this paper.

## Code availability

All analysis codes are available on GitHub at github.com/remsing-group/ePaddleWheel.

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

## Acknowledgements

We acknowledge the Office of Advanced Research Computing (OARC) at Rutgers, The State University of New Jersey for providing access to the Amarel cluster and associated research computing resources that have contributed to the results reported here.

## Author contributions

R.C.R. designed the study, H.S.D. and R.S. performed the simulations, H.S.D. wrote analysis code, H.S.D., R.S., and R.C.R. analyzed data, and H.S.D. and R.C.R. wrote the manuscript.

## Competing interests

The authors declare no competing interests.
