## [Peer Review File · Nature Communications]

Electronic paddle-wheels in a solid-state electrolyteREVIEWER COMMENTS

Reviewer #1 (Remarks to the Author):

The article by Dhattarwal et al reports theoretical findings on the well-known ion conductor AgI. The authors employed molecular dynamics (MD) simulations to investigate correlations in the material in its conducting phase. Density functional theory (DFT) is used together with MD. A variety of time and space correlation functions are applied to analyze the dynamic behavior. The authors suggest that a “paddle-wheel effect” facilitates conduction in AgI.

The study is of high quality but several issues remain. Overall, the study appears not be suited for a broad, high-impact journal because of doubts regarding novelty and significance of the findings, summarized as follows:

1. A previous study by Brenner et al (cited as Ref 15) found that dynamic formation of local geometrical motifs was involved in anharmonic lattice dynamics and the ion conduction mechanism of AgI. The present study certainly went far in their analysis, but in essence the interesting coordination effects seen here are reminiscent of what has been discussed already in Ref 15. In addition, the authors describe that “The ionic diffusion mechanism predicted here is analogous to the paddle-wheel motion observed in molecular solid-state electrolytes.” and “This paddle-wheel mechanism has been leveraged to design better molecular electrolytes”. This means that in other materials it has been observed and exploited before as well. Hence, the present results are not entirely new or different from previous findings in the literature.

2. It is unclear to this reviewer what “paddle wheel effect” means and how the data show it is significant. First, it seems which atomic motions qualify as a “paddle wheel” is subjective. Second, the authors write that “This correlation in cation and lone pair dynamics is reflective of electronic paddle-wheels.”, but why and how? To this reviewer it is not clear how classifying specific atomic motions by a subjective criterion enhances physical understanding of ion conduction. Last but not least, even if for the sake of argument one accepts the definition of paddle-wheel motion, the authors seem to find correlations of it with ion conduction but the reviewer fails to recognize causation. This brings questions

about the significance of said effect for the ion-conduction mechanism. Altogether, the significance of the work to enhance physical understanding in the field is rather limited.

3. The study is limited to one, long-studied and very well-known compound. The article does not suggest implications for how the paddle wheel effect can be used to design new materials. Indeed, it has been discussed before in other studies and compounds. Overall, the broad significance of the work for the materials community therefore seems to be rather limited.

Reviewer #2 (Remarks to the Author):

In this work, the authors extend the concept of paddle wheel mechanism from the field of Solid-state superionic conductors to the case of electronic orbitals. Here the rotating anions are anisotropic atomic species rather than molecular tetrahedra. They use the electronic plastic crystal AgI as a test case to illustrate this new idea and perform on-the-fly DFT dynamics to compute structural properties and time correlation functions. They conclude that Ag⁺ mobility is diffusive as opposed to hopping-like as seen in molecular paddle wheel systems.

They suggest that the idea of electronic paddle wheel can be used as a platform for the rational design of new superionic conductors. The work is a good quality and the results are convincing. This is an important contribution and will be of broad interest.

The authors should address the following points:

1) In the Methods section they authors state “For the calculation of dynamic properties, equations of motion were propagated for 80 ps in the microcanonical ensemble using velocity Verlet integrator with a timestep of 0.5 fs. The last 40 ps of these trajectories were used for the analysis”. Is only one NVE trajectory calculated and time averaged to obtain 8ps of dynamics as shown in Figs. 1d and 1e? Or are many NVE trajectories averaged to obtain the averaging brackets of Eq. (1)? This should be clarified either in the methods section or the Supplementary Information.

2) The authors use an AgI unit cell with edge length of 5.0855 Å. Is this the experimental equilibrium density ($P=0$). Does the DFT model have a similar minimum in the equation of state? ie does the choice of N and V lead to an internal energy minimum as a function of N/V (density= N/V)? Is this independent of temperature?

3) Going from 750K to 100K is a big jump. What kind of phase transition should occur between those 2 temperatures? The authors should mention the superionic transition and its temperature.

minor point:

The authors repeat the word paddle-wheel several times in the last 3 paragraphs before the Acknowledgements. I suggest that they revise the text for a better flow.

Reviewer #3 (Remarks to the Author):

This paper is about new idea "electronic" paddle-wheel effect in solid ionic conductors. The paddle-wheel effect between a diffusing ion and a molecular ion is recently known to increase ionic conductivity in solids. The authors assign lone pairs as a part of paddle-wheel. Atomic "rotation" is described by this paddle-wheel. The idea of electronic paddle-wheel comes from the authors previous work on lone-pair dynamics [17,18].

This new idea is very interesting not only from a scientific point of view but also application to solid-state battery in which high solid ionic-conductor is desired. The authors well documented this effect with FPMD results of AgI super ionic conductor. This paper is publishable in the present form.

Definition of lone pair by Wannier function of I⁻ will work well when I⁻ is interacted to Ag⁺ through the lone pair. If I⁻ is free or weakly bounded, lone pair position may not be defined. But, it is possible to see the correlation between the lone pair and Ag⁺ movement as the authors did. We hope more quantitative analysis; how the correlation improves the conductivity or decrease it, how electronic structure of the lone pair affect it. Anyway, the "electronic" paddle-wheel will be useful to analyze

mechanism of ion diffusion in solids.

The authors showed the collective motion of Ag^+ through the lone pair motion qualitatively. We hope quantitative analysis for the collective motion between the different atoms using the lone-pair dynamics.

A "color" problem was found in lone pair MLWFC in FIG. 1. It is described as "(small, light purple spheres)" in the caption, but color is simply grey on my PC. It may depend on display or pdf.

Reviewer #1:

The article by Dhattarwal et al reports theoretical findings on the well-known ion conductor AgI. The authors employed molecular dynamics (MD) simulations to investigate correlations in the material in its conducting phase. Density functional theory (DFT) is used together with MD. A variety of time and space correlation functions are applied to analyze the dynamic behavior. The authors suggest that a “paddle-wheel effect” facilitates conduction in AgI.

The study is of high quality but several issues remain. Overall, the study appears not be suited for a broad, high-impact journal because of doubts regarding novelty and significance of the findings, summarized as follows:

We thank the reviewer for appreciating the quality of our work and for their comments. We have tried to address all of reviewer’s concerns in the revised manuscript.

1. A previous study by Brenner et al (cited as Ref 15) found that dynamic formation of local geometrical motifs was involved in anharmonic lattice dynamics and the ion conduction mechanism of AgI. The present study certainly went far in their analysis, but in essence the interesting coordination effects seen here are reminiscent of what has been discussed already in Ref 15.

The local geometrical motifs discussed in Ref 15 involve only nuclear degrees of freedom, whereas we focus on the role of *electronic* degrees of freedom in ionic conductivity. The paddle-wheel motion in molecular solid-state electrolytes is a result of molecular rotations. Our work introduces the concept of *electronic paddle-wheels* with the hope of adapting the current understanding of molecular ions to monatomic ions with localized electron pairs that rotate.

In our revision, we have tried to emphasize these points further. We also tried to connect further to Ref 15. We highlight that those previous studies focused on lattice vibrations (nuclear relaxations). Our proposed mechanism adds to this picture by including the role of electronic rotations. Vibrational relaxation is part of the picture and weakens some of the Ag-I interactions, and we also suggest that the diffusion of silver ions is enhanced by the rotation of iodide ions. The missing piece of electronic degrees of freedom is even reflected in Ref 15, where silver ions diffuse (although slower) even after fixing the iodide ions at the BCC lattice position. In that case, electronic paddle-wheel rotations still accompany the diffusion of silver ions in the absence of lattice relaxations.

“The vibrational relaxations in I⁻ sublattice make Ag⁺ more susceptible to diffusion¹⁵. The diffusion, however, is enhanced by the I⁻ lone pair rotations.”

In addition, the authors describe that “The ionic diffusion mechanism predicted here is analogous to the paddle-wheel motion observed in molecular solid-state electrolytes.” and “This paddle-wheel mechanism has been leveraged to design better molecular electrolytes”.

This means that in other materials it has been observed and exploited before as well. Hence, the present results are not entirely new or different from previous findings in the literature.

We emphasize that the paddle wheel in molecular solid-state electrolytes arises from the rotations of molecular ions, a well-understood mechanism that has been exploited to design better molecular electrolytes. However, molecular rotations are absent in monatomic electrolytes. Our study leverages the computational capability to quantify the electronic degrees of freedom and introduces the completely new concept of **electronic** paddle-wheels in monatomic electrolytes. Despite the concept of electronic paddle-wheels being completely new, it has the advantage of analogies with molecular electrolytes, such that years of understanding molecular electrolytes might be leveraged to better understand monatomic ion conductors. We also note that a one-to-one correspondence in the behavior of molecular and electronic paddle-wheels does not always hold. As an example, we showed that ionic conduction with electronic paddle-wheels is diffusive, while a jump mechanism usually operates with molecular paddle-wheels.

2. It is unclear to this reviewer what “paddle wheel effect” means and how the data show it is significant. First, it seems which atomic motions qualify as a “paddle wheel” is subjective.

The coupling between the rotational motion in the host lattice and ionic diffusion, termed the paddle-wheel effect, is a well-established mechanism to explain high ionic conductivity in molecular solid electrolytes like Na_3PS_4 or LiSO_4 (discussed, for example, in the introduction section of Ref 15). The electronic paddle-wheel mechanism, on the other hand, refers to the coupling of electronic rotations and ion diffusion. In the case of AgI, this corresponds to the rotation of iodide electron pairs accompanying the diffusion of silver cations. We have tried to define “paddle-wheel” and “electronic paddle-wheel” more clearly in the revision,

“This phenomenon of the rotational motion of a molecular ion increasing the conductivity of another ion in orientationally disordered molecular solids is termed the “paddle-wheel” effect. The reorientation of molecular ions leads to fluctuations in the local potential that affect the diffusion of mobile ions¹³.”

and

“As summarized by the simulation snapshots in Fig. 1a, electronic paddle-wheels in AgI are defined by the rotation of I^- lone pairs (light purple) facilitating the motion of an Ag^+ cation (orange) from one site to another.”

Second, the authors write that “This correlation in cation and lone pair dynamics is reflective of electronic paddle-wheels.”, but why and how? To this reviewer it is not clear how classifying specific atomic motions by a subjective criterion enhances physical understanding of ion conduction. Last but not least, even if for the sake of argument one accepts the definition of paddle-wheel motion, the authors seem to find correlations of it with ion conduction but the reviewer fails to recognize causation. This brings questions about the significance of said effect

for the ion-conduction mechanism. Altogether, the significance of the work to enhance physical understanding in the field is rather limited.

We respectfully disagree with the statement that our results are subjective. In addition to visually observing electronic paddle-wheel motions in trajectories, e.g. Fig 1a, our correlation functions objectively quantify the dynamics in electronic paddle-wheels. In addition, we also quantify the strong association between the lone pairs of anion and cations (reflected in the joint radial and angular density plots, Fig 1b and 1c) that further suggest a coupling of lone pairs and silver ions.

Indeed, it is difficult to make statements about causation. However, our results clearly show that electronic paddle-wheels exist in the superionic state, while they do not exist in non-conducting states. In molecular paddle-wheels, the same behavior is found (for nuclear degrees of freedom). For diffusion to occur, the paddle-wheel rotations must also occur, and diffusion is switched off if the rotations are switched off. This is true for molecular and electronic paddle-wheels, but the source of the rotations differs between the two (molecular bonds vs. electron pairs).

3. The study is limited to one, long-studied and very well-known compound. The article does not suggest implications for how the paddle wheel effect can be used to design new materials. Indeed, it has been discussed before in other studies and compounds. Overall, the broad significance of the work for the materials community therefore seems to be rather limited.

Indeed, we focused on the very well-known compound AgI to show that the electronic paddle-wheel perspective can give new insights even into AgI. We note that *electronic* paddle-wheels have not been described before, this is the first time, but the analogy with molecular paddle-wheels is beneficial to leverage previous work in that field. We have added a few sentences elaborating on our expectations of how our perspective might aid in materials design.

“These insights help move one step closer to a unified framework for understanding the ion transport in monatomic SSICs in line with molecular liquid and solid electrolytes¹⁷.”

“As a result, we expect that hidden rotational electronic disorder and the resulting electronic paddle-wheels may be an important marker for designing electrolytes through tuning lone pair-mobile ion interactions, for example. We expect that electronic paddle-wheels are not limited to AgI but will be found in many SSICs. For example, while this work was in revision, a preprint appeared that suggested that tin ion lone pair rotations may impact the diffusion of fluoride ions in BaSnF₄²⁸, further emphasizing the importance of understanding electronic effects in SSICs.”

In the above, we also noted that another preprint appeared in the past month that used similar ideas to understand other electrolytes, which emphasizes the significance of our work in understanding solid-state electrolytes.

Reviewer: 2

In this work, the authors extend the concept of paddle wheel mechanism from the field of Solid-state superionic conductors to the case of electronic orbitals. Here the rotating anions are anisotropic atomic species rather than molecular tetrahedra. They use the electronic plastic crystal AgI as a test case to illustrate this new idea and perform on-the-fly DFT dynamics to compute structural properties and time correlation functions. They conclude that Ag⁺ mobility is diffusive as opposed to hopping-like as seen in molecular paddle wheel systems.

They suggest that the idea of electronic paddle wheel can be used as a platform for the rational design of new superionic conductors. The work is a good quality and the results are convincing. This is an important contribution and will be of broad interest.

The authors should address the following points:

1) In the Methods section they authors state “For the calculation of dynamic properties, equations of motion were propagated for 80 ps in the microcanonical ensemble using velocity Verlet integrator with a timestep of 0.5 fs. The last 40 ps of these trajectories were used for the analysis“. Is only one NVE trajectory calculated and time averaged to obtain 8ps of dynamics as shown in Figs. 1d and 1e? Or are many NVE trajectories averaged to obtain the averaging brackets of Eq. (1)? This should be clarified either in the methods section or the Supplementary Information.

We use one long trajectory to calculate all quantities. Time correlation functions are calculated using a windowing approach, where the functions are computed for many possible time origins ($t=0$). This is a standard approach for computing dynamic quantities from equilibrium simulations. To clarify, we added the following sentence to the methods section,

“Calculations for dynamic properties are averaged over multiple time origins.”

2) The authors use an AgI unit cell with edge length of 5.0855 Å. Is this the experimental equilibrium density ($P=0$). Does the DFT model have a similar minimum in the equation of state? ie does the choice of N and V lead to an internal energy minimum as a function of N/V (density= N/V)? Is this independent of temperature?

The unit cell with an edge length of 5.0855 Å corresponds to the experimental equilibrium density. We checked the cell relaxation in the DFT model, and it resulted in an approximately 2% change in the volume at minimum internal energy. However, there is no significant change in the other properties computed from the trajectory obtained from this relaxed cell. Therefore, to be consistent with simulations at different temperatures and to address the application of electrolytes in battery devices (at constant volume), we used a fixed unit cell with the experimental density. Regarding temperature, the experimental structure factor does not

change much with temperature (for a fixed phase), suggesting that our constant volume approximation is reasonable.

3) Going from 750K to 100K is a big jump. What kind of phase transition should occur between those 2 temperatures? The authors should mention the superionic transition and its temperature.

AgI exists in the conducting α phase at higher temperatures and in the non-conducting β (stable) or γ (metastable) phase below ~ 420 K. We have simulated alpha α -AgI at 750 K and the supercooled α -AgI at 100 K, which shows slower iodide ions rotations and silver ions diffusion. In the β -AgI the conductivity will be even slower than α -AgI at 100 K.

We have now mentioned the superionic transition temperature in the discussion of temperature dependence, making it clear that our 100 K results are for a supercooled system,

“In α -AgI at 750 K, above the superionic transition at 420 K⁶, the rotational TCF $C_2(t)$ decays with a relaxation time of approximately 0.4-0.5ps, indicating that MLWFC rotational motion occurs on a picosecond timescale, Fig. 1d.”

and

“When supercooled to the low temperature of 100 K, where α -AgI is no longer conductive (see SI), lone pair rotational dynamics are significantly slowed and α -AgI is no longer in an electronic plastic crystal phase.”

minor point:

The authors repeat the word paddle-wheel several times in the last 3 paragraphs before the Acknowledgements. I suggest that they revise the text for a better flow.

We agree that “paddle-wheel” appears several times, but part of the reason for this is that we are comparing and contrasting “electronic paddle-wheels” and “molecular paddle-wheels.” We have revised slightly to try to improve the readability.

Reviewer #3 (Remarks to the Author):

This paper is about new idea "electronic" paddle-wheel effect in solid ionic conductors. The paddle-wheel effect between a diffusing ion and a molecular ion is recently known to increase ionic conductivity in solids. The authors assign lone pairs as a part of paddle-wheel. Atomic "rotation" is described by this paddle-wheel. The idea of electronic paddle-wheel comes from the authors previous work on lone-pair dynamics [17,18].

This new idea is very interesting not only from a scientific point of view but also application to solid-state battery in which high solid ionic-conductor is desired. The authors well documented

this effect with FPMD results of AgI super ionic conductor. This paper is publishable in the present form.

We thank the reviewer for their time and for the favorable review of our work.

Definition of lone pair by Wannier function of I- will work well when I- is interacted to Ag+ through the lone pair. If I- is free or weakly bounded, lone pair position may not be defined. But, it is possible to see the correlation between the lone pair and Ag+ movement as the authors did. We hope more quantitative analysis; how the correlation improves the conductivity or decrease it, how electronic structure of the lone pair affect it. Anyway, the "electronic" paddle-wheel will be useful to analyze mechanism of ion diffusion in solids.

We thank the reviewer for these suggestions. Indeed, examining how the electronic structure of the lone pairs impacts conductivity is something we are currently working on, but will studying exploring many systems and is beyond the scope of the current manuscript.

The authors showed the collective motion of Ag+ through the lone pair motion qualitatively. We hope quantitative analysis for the collective motion between the different atoms using the lone-pair dynamics.

We do provide quantitative results for the motion in the paper. For example, in Figure 1d,e we show that lone pair rotations and movement of Ag+ occur on the same timescale of ~0.5 ps.

A "color" problem was found in lone pair MLWFC in FIG. 1. It is described as "(small, light purple spheres)" in the caption, but color is simply grey on my PC. It may depend on display or pdf.

We will work with the journal to ensure that the colors can be distinguished. Thank you very much for bringing up this important issue.

REVIEWERS' COMMENTS

Reviewer #1 (Remarks to the Author):

The authors have successfully and thoroughly addressed all comments by this reviewer. No further comments remain.

Reviewer #2 (Remarks to the Author):

The authors have addressed the important concerns in this new version and the paper should be accepted.

Reviewer #1 (Remarks to the Author):

The authors have successfully and thoroughly addressed all comments by this reviewer. No further comments remain.

We thank the reviewer for their time and for the favorable review of our work.

Reviewer #2 (Remarks to the Author):

The authors have addressed the important concerns in this new version and the paper should be accepted.

We thank the reviewer for their time and for the favorable review of our work.